# Exploring a Material-Focused Design Methodology: An Innovative Approach to Pressure Vessel Design

**Edgar Adhair Montes Gómez** [1,†] , **Samantha Ixtepan Osorio** [1,†] , **Luis Arturo Soriano** [2,*] ,
**Guadalupe Juliana Gutiérrez Paredes** [1] and **José de Jesús Rubio** [1]

[1] Sección de Estudios de Posgrado e Investigación, ESIME Azcapotzalco, Instituto Politécnico Nacional, Avenida de las Granjas No. 682, Ciudad de México 02250, Mexico; montesg1700@alumno.ipn.mx (E.A.M.G.); sixtepano@alumno.ipn.mx (S.I.O.); ggutierrez@ipn.mx (G.J.G.P.); rubio.josedejesus@gmail.com (J.d.J.R.Á.)

[2] Escuela Superior de Ingenieria Mecánica y Eléctrica Unidad Azcapotzalco, Instituto Politécnico Nacional, Avenida de las Granjas No. 682, Ciudad de México 02250, Mexico

* Correspondence: lsorianoa@ipn.mx

† These authors contributed equally to this work.

**Abstract:** The design of components and elements comprising industrial machinery, as well as those that are part of an industrial system, has been carried out in recent years using various design methodologies. Currently, the products demanded by customers, as well as the needs of different companies, governments, and individuals, require considerations beyond traditional design, including multidisciplinary aspects such as sustainability, environmental friendliness, and circular economy. The design methodologies considered for this study are the quality function deployment (QFD) methodology, the theory of inventive problem-solving methodology, Ashby's Materials Selection methodology, and the systematic approach methodology, which are currently the main design methodologies. These methodologies present some disadvantages, such as multidisciplinary requirements not being considered directly, the selection of materials based on standards is not well established, and obtaining technical requirements is ambiguous for the technical purposes of the design or manufacturing, and the designer's experience in these examples is important to the design process and the development of the product. For these reasons, the traditional design methodologies are presented, next, a new design methodology is proposed and described, then a case study is performed in order to compare the proposed methodology with traditional design methodologies. Finally, the results show advantages over the traditional design methodologies.

**Keywords:** methodology; pressure vessels; PBSA; design

## 1. Introduction

Currently, one of the processes within engineering is the design of tools, devices, and equipment necessary for energy transformation and utilization. Innovation in design tools or methodologies should embrace sustainable design. Sustainability seeks human well-being and the improvement of quality of life without destroying ecosystems, aiming to relate the dimensions of society, the economy, and ecology for general well-being. In this way, the responsible use of natural resources and the economic and technical feasibility are sought, as well as innovation and development of products with a long lifecycle for highly corrosive processes or those with very low living standards, using materials with better properties to reduce the amount of materials used [1,2].

There are very clear examples of devices that must be properly designed due to the stresses to which they can be subjected in their application and use, such as gears, which are meant for the transformation and utilization of mechanical energy, steam generators for the transformation of thermal energy into electrical energy, the design of vehicles and trains for the transportation of people, the design of electrical devices to enhance home comfort, the design of airplanes or spacecraft, the design of robots for hazardous tasks

in the industry, and recently for medical assistance in patients, the design of medical use devices, etc.

Nowadays, these devices are designed using methodologies that use basic concepts such as fundamental laws or working principles, such as Newton's second law, Archimedes' principle, etc. Product design can be categorized as a plan [3,4], as it involves the search and implementation of useful tools to achieve customer satisfaction, through the gathering and refinement of requirements proposed by the user or customer. There are different methodologies for product design and development, including the TRIZ methodology (theory for inventive problem-solving), APQP Methodology (advanced product quality planning), Ashby's material selection methodology, and Pahl and Beitz' systematic approach (PBSA) [5–17]. Each of these methodologies is designed in a general way for various areas of knowledge, and when a specific device, for example, pressure vessels, is designed, each of them presents advantages and disadvantages. Before selecting any of the aforementioned methodologies, an analysis of the situation must be conducted to ensure obtaining a viable solution for design and innovation. Therefore, each of the existing methodologies for the design of pressure vessels is described in a general way below.

In the QFD methodology, the advantages of this approach include establishing a relationship between customer requirements and the technical specifications developed during the product development process. Moreover, an analysis of existing market proposals is carried out with the main objective of identifying specifications that can be improved or establishing minimum requirements. However, this methodology also has some disadvantages. For example, some customer requirements can be ambiguous, making it challenging to establish them as technical requirements. As a result, such requirements may not be correctly addressed. Similarly, the use of the house of quality can lead to a lengthy interactive process, as it involves a group of people working together to find a solution.

In the case of the TRIZ methodology, there are also advantages. The first is the use of already established parameters for improving the design to be performed. Secondly, the design is carried out using principles of inventiveness. However, the main disadvantage of this methodology is the challenge of making a well-defined relationship between the inventive processes and the matrix of contradictions. This requires a significant amount of experience from the designer, which is key for the correct execution of this methodology. Ashby's material selection methodology is a framework that is based only on the selection of the material and geometry of the product to be designed. In addition, this methodology is considered complementary and inherent to the manufacturing field since it is tailored to elements that can be produced by a transformation process. On the other hand, the PBSA methodology is a methodology based on a conceptualization of the design that is demanded by the customers. The search for principles of operation marks the beginning of product design, which tends to be a lengthy process requiring at least a minimal experience with the type of products to be designed. This methodology tries to avoid iterative processes by trying to define the design conditions from the beginning, although this step is the most delayed in terms of the search for the restrictions.

The design of a chemical reactor requires considering the combination of both pyrolysis and gasification processes; these types of processes are carried out at high internal pressures and temperatures in a medium filled with tars resulting from the thermochemical transformations of the process. The design of pressure vessels is carried out following established codes or standards, such as the ASME Code (American Society of Mechanical Engineers) Section VIII, EN 13,445, AD Merkblatter, among others [17]. Standards are used for the manufacturing, design, and maintenance of pressure vessels because they are designed to prevent accidents within the workplace [17]. Reactors are devices in the chemical industry used to carry out chemical reactions on a small and large scale. These devices can be designed following the ASME Code standards for pressure vessels or the TEMA (Tubular Exchanger Manufacturers Association) for heat exchangers [18,19]. These standards provide safety factors for the design of internal pressure, heat transfer, and process efficiency.

For example, the scientific community has made efforts to propose and implement mixed methodologies. Aydin et al. [20] developed a decision-making process for planning activities using a mixed methodology in the Fuzzy environment and the QFD methodology. Chen et al. [21] performed a mixed methodology between the QFD and the fuzzy environment to value technical attributes that can be developed in a product. Chen et al. [21] developed a flexible manufacturing system to give value to customer requirements. Kulcsár et al. [22] complemented the QFD methodology with network science for the evaluation of alternatives to determine product development objectives. Wang et al. [23] developed an algorithm for the integration of QFD and TRIZ methodology for innovative product development, using the house of quality to find customer requirements and use inventive principles to solve them.

Thus, traditional design methodologies often do not consider safety features or atypical conditions during the design process in order to determine the most suitable design. For these reasons, multidisciplinary design methodologies are gaining attention, and their development and proposal are becoming a research topic. Recently, a hybrid methodology for the design of a gasifier was proposed by Ferrer et al. [24]. The main motivation of this work is the design of chemical reactors; the authors propose a combination of the Eco-innovation methodology and TRIZ methodology as a multidisciplinary methodology to design products for the chemical industry.

Finally, this paper is organized as follows: Section 2 presents the quality function deployment (QFD) methodology. In Section 3, the theory of inventive problem-solving methodology is presented. Section 4 covers Ashby's materials selection methodology. Section 5 introduces the systematic approach methodology. In Section 6, a comparison of the aforementioned methodologies is presented. Section 7 describes the material-focused design methodology. Section 8 contains the case study. Section 9 discusses the findings, and Section 10 presents the conclusions.

## 2. QFD Methodology

The quality function deployment (QFD) methodology has been applied to external instruments of a reactor or a pressure vessel, as in the case of Ismail et al. [25] for the development of a deployment arm for a platform in a boiler using this methodology. The quality function deployment methodology is focused on transforming customer desires into technical requirements for product design [26,27]. This methodology is utilized in the development and improvement of activities, enabling its application to manufacturing processes, services, and product design, as in the case of Lorenzo et al. [28], who applied the QFD (quality function deployment) methodology to healthcare management, and Sharma and Rawani [29] in healthcare. Likewise, it has been used as a complementary tool for pressure vessels, like in the case of Ismail et al. [25] for a deployment arm platform. Table 1 shows different applications of the QFD methodology in various areas of study, including design, manufacturing, planning, chemical processes, health, and services.

The QFD methodology can be applied in different branches of study. Among the applications in the area of design are the design of a suction cup end effector by Ramírez Gordillo [30], a ceramic tile by Erdil and Arani [26], various products by Kuys et al. [31], and an airbag by Wang et al. [23]. In product manufacturing, it has been used as a manufacturing planning tool by Crowe and Cheng [32], and for examining customer expectations with CNC machines and technical requirements by Kulcsár et al. [22]. The methodology is also adaptable for process planning, as demonstrated by Yang et al. [33] for performance evaluation in oil and gas plants. For benchmarking purposes, Schillo et al. [34] used the methodology for policy linkage with biofuels. In the service sector, Tamayo Enríquez et al. [15] applied the QFD methodology for sporting events and music concerts, while Chen and Ngai [35] used it for complex product planning.

Similarly, the methodology was used by Partovi [36] for process selection in the chemical industry. In the health sector, Lorenzo et al. [28] applied the QFD methodology for health management, and Sharma and Rawani [29] used it for the product Syringe and

Needle. In the area of process evaluation, Tottie and Lager [37] used the methodology for the evaluation of a process chain, while Aydin et al. [20] used it for the development of sustainable policies in the retailing industry. The relevant works mentioned and their applications are summarized in Table 1.

**Table 1.** Applications of the QFD methodology

| Application | Author | Reference |
|---|---|---|
| Design | [Ramírez Gordillo, Javier] | [30] |
| | [Erdil, Nadiye Ozlem, Arani Omid M.] | [26] |
| | [Kuys, Blair et al.] | [31] |
| | [Wang, Hao et al.] | [23] |
| Manufacturing | [Crowe, Thomas J. Cheng, Chao-Chun] | [32] |
| | [Kulcsár et al.] | [22] |
| | [Gandhinathan et al.] | [38] |
| Planning | [Yang et al.] | [33] |
| | [Cherifâ et al.] | [39] |
| | [Schillo et al.] | [34] |
| | [Tamayo Enríquez et al.] | [15] |
| | [Chen et al.] | [21] |
| Chemistry | [Partovi, Fariborz Y.] | [36] |
| Health | [Lorenzo et al.] | [28] |
| | [Sharma, J. R. Rawani, A. M.] | [29] |
| Services | [Tottie, Magnus, Lager, Thomas] | [37] |
| | [Aydin et al.] | [20] |

The methodology for quality function deployment (QFD) has become a tool for communication between marketing and production processes in the design of new products [27]. The key part of the QFD methodology is based on the house of quality diagram, which consists of different elements for the development of the methodology and the presentation of each of the requirements for the product design to be developed [20,26,28–30]. The house of quality is shown in Figure 1 and it consists of six different matrices. According to [26,28–30], the objective of each matrix is described as follows:

1.  The first matrix identifies customer needs.
2.  The second matrix identifies technical requirements and determines their interrelationships.
3.  The third matrix determines the relationship between customer needs and technical requirements.
4.  The fourth matrix performs a competitive analysis against other existing products.
5.  The fifth matrix identifies the most important requirements and technical difficulties.
6.  The sixth matrix calculates the importance indices.

The methodology process is carried out in four different stages [26], which are explained in Figure 2 as follows:

*   Design.
*   Details.
*   Process.
*   Production.

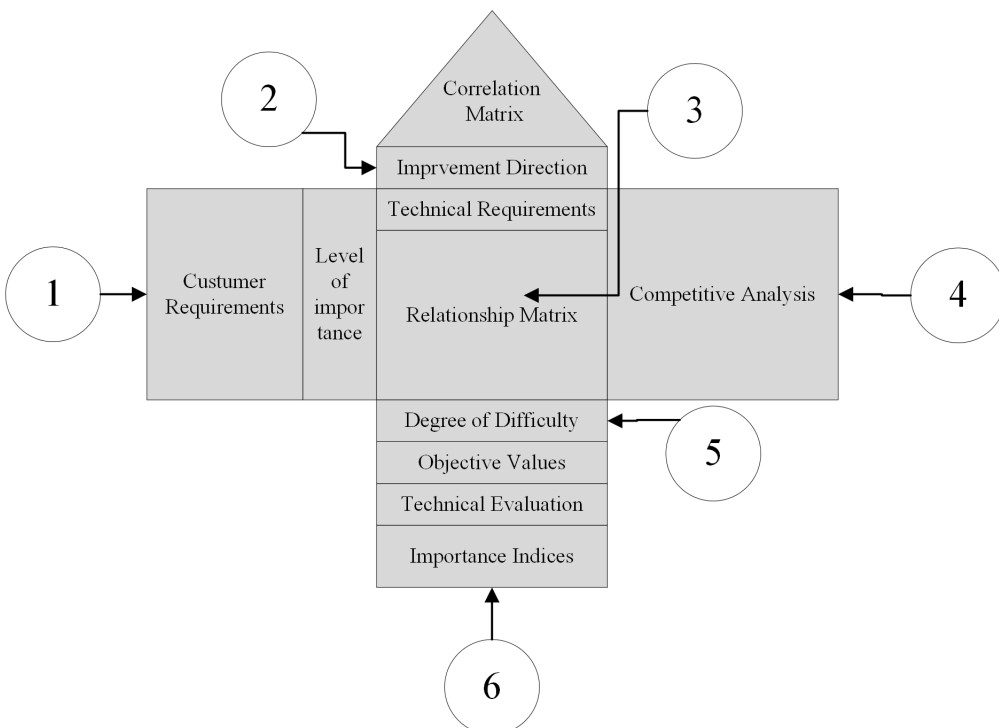

**Figure 1.** House of quality in the QFD methodology [20,26,39].

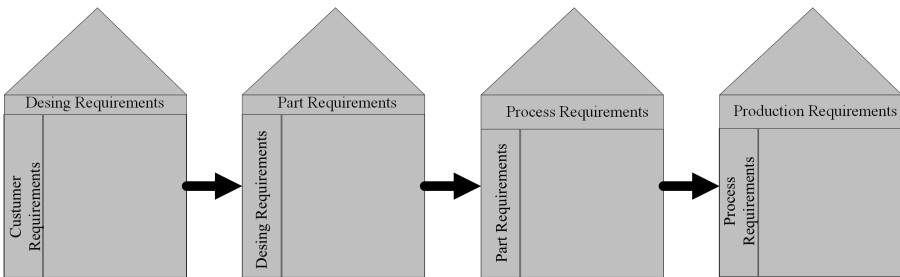

**Figure 2.** Quality function deployment methodology process [26].

In the design stage, customer requirements are related to the technical design requirements to obtain the main characteristics and needs that the product should fulfill [26].

In the details stage, technical design requirements are related to the requirements of the parts that will compose the final element to be produced [26].

In the process stage, the characteristics of the parts or part requirements are related to the process requirements [26].

In the production stage, the process requirements are related to the requirements for producing the final element and obtaining the product [26].

To carry out this process, it is necessary to create at least one house of quality for each stage of the methodology.

The use of the quality function deployment methodology provides advantages such as the possibility of achieving overall efficiency in terms of improvements or corrections to establish task priorities for obtaining the final product. It promotes teamwork in design, manufacturing, and marketing, as well as collaboration across different areas to obtain the final product.

The methodology also has disadvantages, such as subjectivity in customer needs, which may include irrelevant desires. Additionally, the design process using this methodology takes a lot of time to work together with different areas to obtain the final product, and obtaining each of the customer needs is a time-consuming process.

The use of the quality function deployment methodology in the design of pressure vessels or reactors is relatively problematic because there are no specific opinions or needs that the customer may have regarding the design. However, considerations related more to the use or instrumentation that the product may need are prevalent. Therefore, the use of this methodology is more applicable to peripheral elements, controls of vessels or reactors, and ergonomic functionality for operating these elements.

### 3. TRIZ Methodology

The TRIZ methodology (translated as the theory of inventive problem-solving, or by its Russian acronym, Theoria Resheneyva Isobretatelskehuh Zadach (TRIZ)) [24,40,41] is a methodology that operates through generic models for problem-solving. The basic process of TRIZ is illustrated in Figure 3.

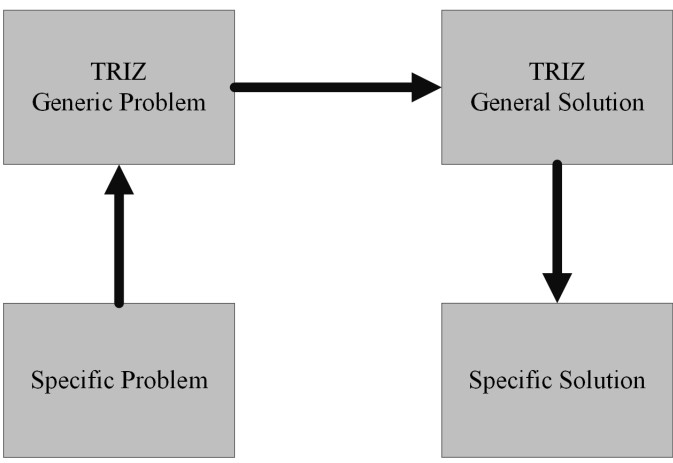

**Figure 3.** Basic process of TRIZ [24,41].

Altshuller, the creator of this methodology [42], conducted observations to analyze invariants in problem-solving during the development of innovations and scientific discoveries [40,42]. This methodology uses the contradiction matrix and inventive principles as design tools. It works with a contradiction matrix, where the inventive principles to be changed are related to finding the inventive principles to work on for the product to be developed. An excerpt from the contradiction matrix can be seen in Table 2, where engineering characteristics to modify are used to find the inventive principles to work on.

The contradiction matrix operates by comparing the engineering characteristics among themselves. By performing this operation, it is possible to determine what type of inventive principles can be modified when wanting to modify some engineering characteristic. For instance, in the case of comparing the length of a movable object against the weight of a movable object, the inventive principles that can be modified are counterweight, dynamicity, pneumatic or hydraulic constructions, and rejecting and regenerating parts [41,42].

With the use of the contradiction matrix, it is possible to determine which types of inventive principles would have to be modified to solve the technical contradiction. It is necessary to understand each inventive principle in order to use them as tools for the design, and together with the experience of the designer, to develop a product that complies with the conditions of the process to be carried out.

**Table 2.** Contradiction matrix.

| Characteristics | | 1 | 2 | 3 | 4 | 5 | 6 | 7 | 8 |
|---|---|---|---|---|---|---|---|---|---|
| | | | | | | **Characteristics** | | | |
| Weight of a movable object | 1 | X | | 15, 8, 29, 34 | | 29, 17, 38, 34 | | 29, 2, 40, 28 | |
| Weight of a stationary object | 2 | | X | | 10, 1, 29, 35 | | 35, 30, 13, 2 | | 5, 35, 14, 2 |
| Length of a movable object | 3 | 8, 15, 29, 34 | | X | | 15, 17, 4 | | 7, 17, 4, 35 | |
| Length of a stationary object | 4 | | 35, 28, 40, 29 | | X | | 17, 7, 10, 40 | | 35, 8, 2, 14 |
| Area of a movable object | 5 | 2, 17, 29, 4 | | 14, 15, 18, 4 | | X | | 7, 14, 17, 4 | |
| Area of a stationary object | 6 | | 30, 2, 14, 18 | | 26, 7, 9, 39 | | X | | |
| Velocity | 7 | 2, 26, 29, 40 | | 1, 7, 4, 35 | | 1, 7, 4, 17 | | X | |
| Force | 8 | | 35, 10, 19, 14 | 19, 14 | 35, 8, 2, 14 | | | | X |

Note: *X* indicates that the characteristic not have relation.

Each inventive principle represents a way to solve typical contradictions by modifying certain aspects to improve, increase, or reduce certain characteristics of the product. These inventive principles are found in Table 3, where the forty inventive principles of the methodology are mentioned. Table 4 also presents different fields in which the TRIZ methodology is used. This methodology has found use in various areas such as the design of chemical reactors, chemical processes, planning, component design, and manufacturing, as well as in education, healthcare, and service sectors.

**Table 3.** Inventive principles of TRIZ [42].

| | Inventive Principle | | Inventive Principle | | Inventive Principle |
|---|---|---|---|---|---|
| 1 | Segmentation | 14 | Spheroidality | 27 | Dispose |
| 2 | Extraction | 15 | Dynamicity | 28 | Replacement of Mechanical System |
| 3 | Local Quality | 16 | Partial or Excessive Action | 29 | Pneumatic or Hydraulic Constructions |
| 4 | Asymmetry | 17 | Transition Into a New Dimension | 30 | Flexible Membranes or Thin Films |
| 5 | Consolidation | 18 | Vibration | 31 | Porous Material |
| 6 | Universality | 19 | Periodic Action | 32 | Changing the color |
| 7 | Nesting | 20 | Continuity of Useful Action | 33 | Homogeneity |
| 8 | Counterweight | 21 | Rushing Through | 34 | Rejecting and Regenerating Parts |
| 9 | Prior Counteraction | 22 | Convert Harm into Benefit | 35 | Transformation of Properties |
| 10 | Prior Action | 23 | Feedback | 36 | Phase Transition |
| 11 | Cushion in Advance | 24 | Mediator | 37 | Thermal Expansion |
| 12 | Equipotentiality | 25 | Self-service | 38 | Accelerated Oxidation |
| 13 | Do it Reverse | 26 | Copying | 39 | Inert Environment |
| | | 40 | Composite Materials | | |

The application of the TRIZ methodology can be performed in different branches of study. Among the applications are reactors, as in Kim et al. [43], who used the TRIZ methodology to improve the security of chemical processes in reactors. Ferrer et al. [24] used a hybrid methodology for the design of a gasification reactor. In the area of chemical processes, Cortes Robles et al. [44] applied the methodology, while Abdul Rahim et al. [45] used it for the development of a new chemical product. Srinivasan and Kraslawski [46] applied the methodology for the design of inherently safer chemical processes. Pokhrel et al. [47] used the methodology for solving problems in process engineering that involve physical and chemical changes. Additionally, Vaneker and Van Diepen [48] utilized it for maintenance task planning. In the education field, Berdonosov [49] applied the methodology for planning, while Lee et al. [50] used it for customized and knowledge-centric service. Khodadadi and Von Buelow [51] combined the TRIZ methodology with a genetic algorithm for the design exploration of a folded plate dome. Rau et al. [40] used the methodology for exploring green product design. Li et al. [52] applied it for a patent review and novel design of a vehicle classification system. Delgado-Maciel et al. [53] used the methodology for the evaluation of conceptual design, and Munje et al. [54] applied it to the design of a CPU fan in additive manufacturing.

One of the main advantages of using the TRIZ methodology is its reliance on the analysis of patents and inventive processes, making it a valuable tool for generating new products in the industry. It is a fast process due to the tools used within the methodology. However, this poses a challenge due to the knowledge or training required to apply these methodologies effectively to processes.

**Table 4.** Applications of the TRIZ methodology.

| Application | Author | Reference |
| --- | --- | --- |
| Reactors | [Kim et al.] | [43] |
| | [Ferrer et al.] | [24] |
| Chemical processes | [Cortes et al.] | [44] |
| | [Abdul Rahim et al.] | [45] |
| | [Srinivasan, Kraslawski] | [46] |
| | [Pokhrel et al.] | [47] |
| Planning | [Vaneker, Van Diepen] | [48] |
| | [Berdonosov, Victor] | [49] |
| | [Lee et al.] | [50] |
| Design | [Khodadadi. Von Buelow] | [51] |
| | [Rau, Hsin, Wu, Katrina Mae] | [40] |
| | [Li et al.] | [52] |
| | [Delgado-Maciel et al.] | [53] |
| Manufacturing | [Munje et al.] | [54] |

## 4. Materials Selection Methodology

One of the methodologies applied to the design of mechanical elements, such as structures, mechanisms, and gears, as well as for tool design and the selection of materials for different tools or mechanisms, is Ashby's materials selection methodology. Consequently, this methodology can be used for selecting materials for pressure vessels and reactors, taking into account the working conditions in which they will be used. Ashby's materials selection methodology concentrates on optimizing the functionality or performance of the piece to be designed.

This performance can be denoted as follows:

$$P = f[(F), (G), (M)] \tag{1}$$

where *P* represents the performance of the piece to be designed, *F* denotes the functional requirements of the piece to be designed, *G* denotes the geometric properties of the piece to be designed, and *M* represents the properties of the materials [55–57].

The Ashby methodology begins with generating a new idea and starts with different steps to obtain the complete methodology. The steps are developed according to the following diagram in Figure 4:

- Concept: Involves the conceptualization of how the product to be designed will function, determining its structure and the principles of its operation.
- Realization: Represents how the product model will be created, how the assembly will be carried out, product evaluation, and the selection of the most suitable materials.
- Details: Analyzing components and optimizing product performance.

In case of not finding them, iterate as many times as necessary until all the parameters are complied with.

The product to be developed is considered a technical system consisting of components and subsystems.

The use of the Ashby methodology focuses on material selection for equipment requirements. This selection is made by translating the equipment requirements into properties of the material to be used Figure 5.

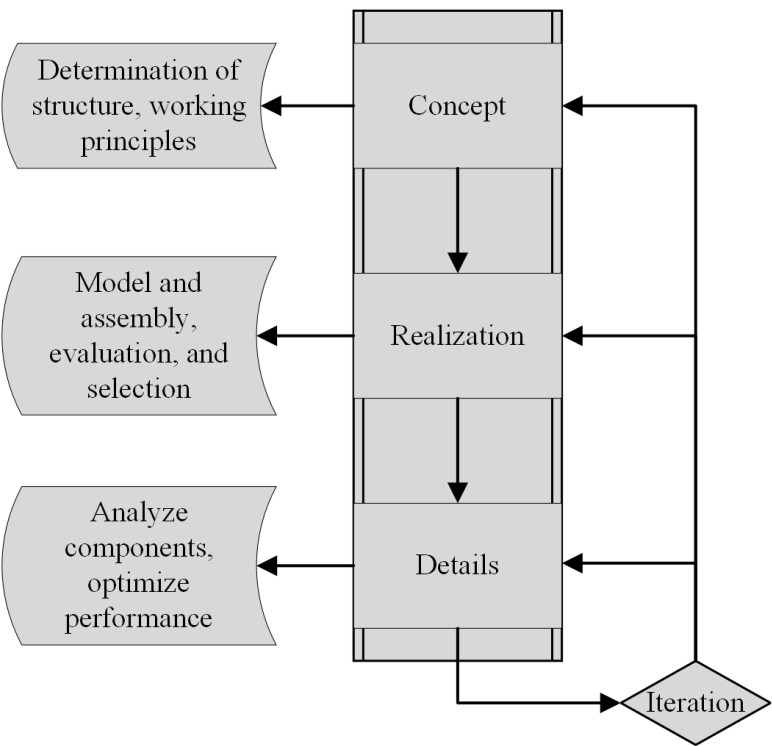

**Figure 4.** Steps of the Ashby methodology [58].

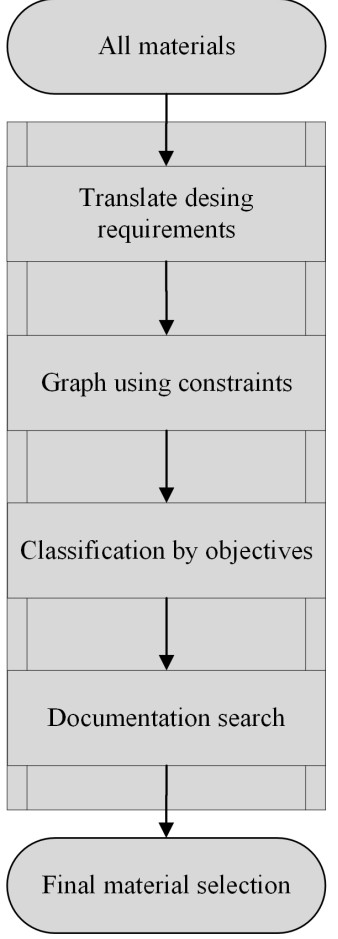

**Figure 5.** Materials selection procedure [55].

From Figure 5, this selection is conducted by following the steps below:

- Translate design requirements expressed as functions, constraints, objectives, and free variables that could be used in the product.
- Create a graph with the restrictions to eliminate materials that do not meet the specifications of the process to be developed.
- Classify by objectives, such as certain ranges of values in material properties, and find the materials that best fit the constraints of the working system.
- Search for documentation of the selected material, including family history or candidates best suited for the process to be carried out.

To finally find the material choice, the process can be conducted as follows, ultimately obtaining the material that meets the requirements or properties suitable for the equipment to be designed.

In the Ashby methodology, parameters known as material indices are identified [59]. These indices are related according to the physical characteristics of the selected materials, such as material density, modulus of elasticity, hardness, thermal conductivity, specific heat, electrical resistivity, etc. Material properties can be classified according to Table 5.

**Table 5.** Material indices [60].

| Class | Property | Symbol and Units |
|---|---|---|
| General | Density | $\rho\ [\frac{\text{kg}}{\text{m}^3}]$ |
| | Price | $C_m\ [\frac{\$}{\text{kg}}]$ |
| Mechanical | Elastic Modulus | $E, G, K$ [GPa] |
| | Poisson's ratio | $v$ |
| | Failure strength | $\sigma_f$ [MPa] |
| | Fatigue strength | $\sigma_e$ [MPa] |
| | Hardness | $H$ |
| | Fracture toughness | $K_{1c}$ [MPa m$^{1/2}$] |
| | Loss coefficient | $\eta$ |
| Thermal | Thermal conductivity | $\lambda\ [\frac{\text{W}}{\text{mK}}]$ |
| | Thermal diffusivity | $a\ [\frac{\text{m}^2}{\text{s}}]$ |
| | Specific heat | $C_p\ [\frac{\text{J}}{\text{kgK}}]$ |
| | Coefficient of thermal expansion | $\alpha\ [^o\text{K}^{-1}]$ |
| Electrical | Electrical resistivity | $\rho_e$ [$\mu\ \Omega$ cm] |

The value of each material property will change according to the type of materials analyzed. This variability allows for the graphing and restriction of materials with the proposed conditions, enabling the correlation of these properties to obtain behavior graphs and analyze which materials are suitable for the proposed conditions.

The use of the indices is done by correlating the properties of the materials against each other, depending on the behavior to be analyzed. Using Young's Modulus against the density of the material provides insight into the behavior of limited rigidity and minimum mass.

All applications of Ashby's material selection methodology are used in product design and validation, using various material indices, such as thermal conductivity with electrical conductivity [61], bandgap with dielectric constant [62], Young's Modulus with energy content [63], and Young's Modulus with atmospheric pollution index [63]. In all cases, each index shown identifies a series of materials that can be used depending on the specified needs.

Table 6 presents the different applications for the use of Ashby's material selection methodology. These applications utilize certain indices for the correct selection of materials, depending on the constraints of the process being considered.

**Table 6.** Applications of Ashby's materials selection methodology.

| Application | Author | Reference |
| --- | --- | --- |
| Bipolar plates for polymer electrolyte | [De Oliveira et al.] | [61] |
| Thermal management of a car cabin | [Das et al.] | [64] |
| Microelectronic heat sinks | [Prashant Reddy, Gupta] | [65] |
| Car body stampings | [Antunes, De Oliveira] | [66] |
| Semiconductors | [Aditya, Gupta] | [62] |
| Buildings | [Beltran et al.] | [67] |
| Biocomposites | [Shah, Darshil U.] | [68] |
| Nuclear applications | [Moschetti et al.] | [69] |
| Aerospace | [Ahmad et al.] [Yavuz] | [70] [71] |
| Biomass combustion | [Antunes, De Oliveira] | [72] |
| Micro-electromechanical | [Guisbiers et al.] | [73] |
| Beverage containers | [Holloway] | [63] |
| Sports equipment | [Bird et al.] | [74] |
| Engines | [Djassemi, Manocher] | [75] |

## 5. Methodology of the Systematic Approach (PBSA)

The methodology proposed by Pahl and Beitz, known as the "Systematic Approach", describes 'design' in four phases: idea clarification, conceptual design, incorporation design, and detailed design; see Figure 6 [76,77].

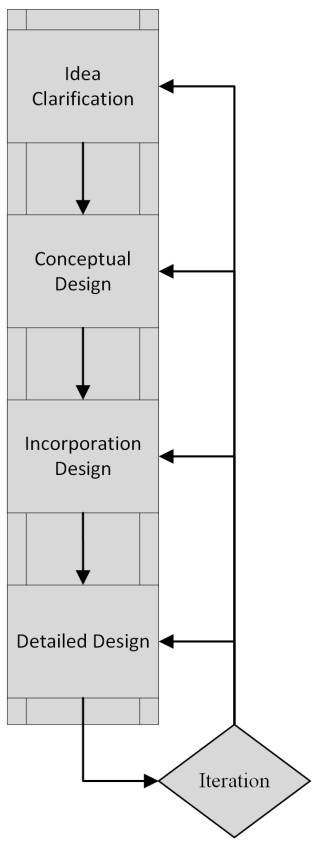

**Figure 6.** Steps of the systematic approach methodology.

The clarification of ideas is based on the collection, search, and documentation of the product requirements to be designed.

Once the requirements are gathered, it is necessary to identify which are demands and which are desires. Requirements identified as demands must be resolved upon completing the design of the element to work on, but desires are taken into consideration during the design of the element. Once the totally necessary requirements are identified, it should be ensured that with these requirements, there is a definition of the product concept, the structure of the product to be designed, and a determination of the overall embodiment of the product made.

All identified requirements must be categorized into partial requirement lists [76].

The conceptual design is carried out by identifying the basic design principles to be applied; these principles are based on the working conditions and physical, thermal, and mechanical phenomena. The operating principles encompass the functioning and composition of the element and the ability to predict its behavior under working conditions or boundary conditions.

The incorporation design is a part of the methodology that begins with the product concept solution and develops it based on the criteria established in the conceptual design.

The detailed design is where the realization design is completed, aiming to complement the forms, dimensions, operating processes, and costs of the final product.

Table 7 shows some of the different applications of the methodology developed by Pahl and Beitz. The use of this methodology has mostly been for the design of mechanical components.

**Table 7.** Applications of the systematic approach methodology by Pahl and Beitz.

| Application | Author | Reference |
| --- | --- | --- |
| Gearbox (Gears) | [Fiorineschi et al.] | [78] |
| Prosthesis | [Lelieveld, Maeno] | [79] |
| Automated test systems | [Mendes et al.] | [80] |
| Design process | [Kamarudin et al.] | [81] |
| Industrial robot | [Ore et al.] | [82] |

The application of Pahl and Beitz's systematic approach methodology is utilized in various areas of device design. For example, Fiorineschi et al. [78] applied it to the design of a gearbox for concrete mixers. Lelieveld and Maeno [79] used the methodology in the design of prostheses. Mendes et al. [80] used the methodology for designing automated test systems. Kamarudin et al. [81] used it for modeling the conceptual design process. Additionally, Ore et al. [82] used the methodology for the design of human-industrial robots. All these applications follow the systematic approach process to achieve a product design.

## 6. Methodology Comparison

In order to make a decision regarding the use of a design methodology for product development, Table 8 shows an analysis of the advantages and disadvantages of each traditional design methodology. As a result of this analysis, the criteria used to choose a methodology are as follows:

- Technical parameters for the design to be carried out.
  If the methodology considers technical parameters for the product to be developed, such as hardness, rigidity, fluids to be used, pressures, temperatures, etc.
- Search for physical phenomena.
  If the methodology considers what types of physical phenomena or principles may affect the design or on which the design.

- Adaptability to design.
  If it is possible to use the methodology in different design fields, such as health or services.
- Simplicity.
  If the design can be applied in a simple manner or if it is a complicated process.
- Exposure of requirements.
  If the methodology considers requirements that can be obtained from the consumer or client.
- Conceptual.
  If the methodology somehow presents the concept of the design to be resolved.
- Material properties.
  If the methodology considers the physical characteristics of the materials for the design and not just as a detailed process.
- Flowchart.
  If the methodology can be represented in a block flowchart.

**Table 8.** Methodology comparison.

| Conditions | QFD | TRIZ | Ashby | SAPB |
|:---:|:---:|:---:|:---:|:---:|
| Finding technical parameters | X | X | X | X |
| Search for physical phenomena | | X | X | X |
| Adaptability to design | X | X | | X |
| Simplicity | X | | X | |
| Exposure of requirements | X | X | X | X |
| Conceptual | | X | | X |
| Material properties | | | X | |
| Flowchart | X | | X | X |
| Quantification. | 5 | 5 | 6 | 6 |

Note: *X* indicates that is a feature considered by the methodology.

## 7. Material-Focused Design Methodology

The current methodologies for engineering design are conceived to consider general aspects when designing a solution for customers. Consequently, these methodologies sometimes struggle to adapt to product development when the proposed solution requires consideration of specific requirements, particularly in the context of mechanical design. For instance, the design of energy transformation equipment, such as internal combustion engines, heat exchangers, and boilers, is based on principles like the laws of thermodynamics, Fourier's law, heat and mass transfer phenomena, and Archimedes' law. Additionally, for stress loads, the type of process being performed is a determining factor. On the other hand, devices such as gears and mechanisms are designed under motion transfer laws, including Newton's laws. Each of these devices could be subjected to various operational conditions like high temperatures, dilatation, shear stresses, internal or external pressures, impacts, and so forth. The operating conditions play a crucial role in the selection of materials and consideration of safety factors by the designer. If this process is not carried out according to standards, the proposed solution in mechanical design might exhibit errors or premature failures. Therefore, the selection of materials can become a recursive and challenging process for designers. Selecting a methodology that includes all features of materials and client requirements could be a good solution for them; however, the current methodologies are incomplete to satisfy all these requirements.

Traditional methodologies offer advantages; for instance, TRIZ requires the designer's experience in interpreting the contradiction matrix and in developing inventive principles to create an adequate solution. If the designer has extensive experience in product design, their proposal is likely to be highly accepted by the client because it can satisfy their

requirements. Similarly, in the QFD methodology, the designer's experience is crucial as they transform client descriptions into technical requirements. This process is a hard task if the designer does not have experience. Flowcharts in these methodologies are essential for following the necessary steps, but they are not entirely sufficient as they do not always incorporate standards into product design.

Although traditional methodologies have their merits, it is necessary to develop a methodology that considers the majority of features and standards to create solutions that satisfy client requirements. Recently, the development of new methodologies has gained attention as an option for product design. Aydin et al. [20] proposed an interesting methodology for product design, which includes decision-making for activity planning. Chen et al. [21] designed a mixed methodology combining QFD with the fuzzy environment for evaluating technical requirements, which could help determine the most important requirements for product design. Kulcsár et al. [22] enhanced the QFD methodology with network science for evaluating alternatives to determine the objectives of the design to be developed. Wang et al. [23] developed an algorithm to integrate QFD and TRIZ methodology. In the context of multidisciplinary design, all these advances could be applied. However, the design process becomes iterative and time-consuming compared to its application in other fields, mainly because these methodologies, despite using flowcharts and providing a step-by-step guide, do not consistently use standards commonly applied in design.

1.  Clarification of ideas.
    Identify the needs, desires, and requirements that can be applied to the final product design. In this way, gather as much information as possible that can be applied to the product.
2.  Technical requirements.
    Filter the requirements collected in the clarification of ideas. The filtering is based on the importance, principles, and subjectivity of the requirements, similarly classifying each requirement and determining its value, depending on its importance.
3.  Conceptual design.
    During the conceptual design, the search for principles that can be applied to the product, as well as the state of the art (if required) for the product. This process includes obtaining the equations governing the process, as well as the laws, methods, and theories that define the process.

    (a) Search for principles.
        This involves searching for all possible documentation that studies the process to be developed, such as governing equations, theories, laws, methods, and codes applicable to the product design.
    (b) Value of principles.
        Classify and value the technical principles depending on the requirements already valued in the previous step.
4.  Prototype design.
    Execute and apply the conceptual design for the construction of the final product prototype, as well as the necessary steps for its manufacturing.

    (a) Translate the requirements.
        Apply each of the higher-value principles to obtain the product, applying the governing equations, laws, methods, and codes for obtaining the final product.
    (b) Graph using restrictions.
        Select the type of material using the graphs and material indices applied in Ashby's materials selection methodology to establish the most suitable material for the final product with the restrictions of the requirements.
    (c) Documentation of materials.
        Obtain the physical properties of the selected materials according to the restrictions of the requirements.

5. Details.

Applied to obtain usage protocols, tolerances, the search for alternative materials, and specifications for manufacturing and mass production.

## 8. Case Study

### 8.1. Problem Statement

The design of a pressure vessel for the pyrolysis process is carried out, considering certain mechanical and thermal conditions of the process. The pyrolysis process is focused on the thermal degradation of biomass, to obtain products such as tars, chars, and syngas.

The pyrolysis process can be modeled using single-reaction kinetics, where the thermal degradation of biomass is proportional to the residual mass at a given time. This modeling utilizes the Arrhenius law, which describes the first stage of biomass degradation with the following equation:

$$Solid \xrightarrow{k} Volatile + Chars \tag{2}$$

The main drawbacks that occur during the pyrolysis process include high temperatures, which range from 300 °C to over 1000 °C at the higher end. The corrosion caused by these high temperatures inside the pressure vessel leads to thermal degradation. Moreover, during the pyrolysis process, a low pressure is generated, requiring the minimum thickness of the pressure vessel to support at least 4 bar and a maximum of 6 bar. The pressure vessels are designed using the ASME code. This standard proposes safety geometry for the vessels and the security system, and recommends the use of relief valves.

This process is centered on the supply of an energy flux to raise the temperature of the container vessel and begin the internal process. The requirements for the product design are shown in the Table 9 as follows:

**Table 9.** Customer requirements.

| Requirement | Value |
| --- | --- |
| Maximum process temperature | 1000 °C |
| Heating rate | 100 $\frac{°C}{min}$ |
| Pyrolytic temperature | 600 °C |
| Maximum working pressure | 6 bar |
| Minimum design pressure | 4 bar |
| High-temperature durable material | List |
| Compliance with international standards | List |
| Safety system | List |
| Low corrosion | List |
| Low vessel costs | List |
| Compliance with geometric dimensions and tolerances | List |

### 8.2. Material-Focused Design Methodology

In order to apply the methodology proposed in Figure 7, a flow diagram outlining the steps to follow is shown. The first step in developing the methodology is to clarify ideas. This is followed by searching for technical requirements, designing the prototype, and finally, detailing the final design.

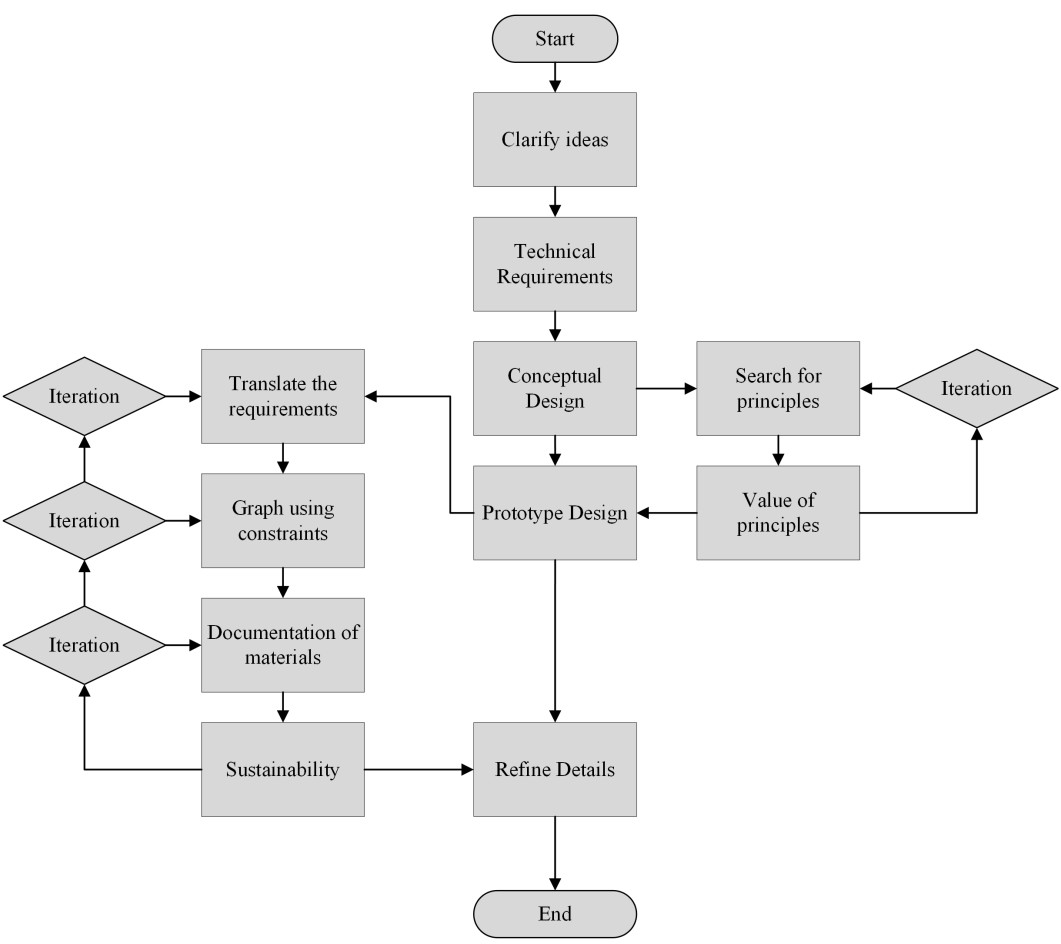

**Figure 7.** Material-focused design methodology diagram.

8.2.1. Clarify Ideas

The clarification of ideas involves finding a clear concept for each client requirement. For instance, in reactor design, the requirements are based not only on the client's desires but also on the main working conditions of the reactor. The pyrolysis process occurs in an environment controlled at 600 °C. To carry out this process, a heat supply of $\frac{°C}{min}$ is required. To maintain the pyrolytic temperature, a continuous supply of biomass, such as sawdust, is necessary. The pyrolysis process takes place inside a reactor, creating a highly corrosive environment. In the reactor, the working pressure is set at 4 bar, making it a vessel subject to internal pressure. For these reasons, the initial working characteristics are considered when beginning the application of the design methodology, and it is shown in Table 10 as follows:

**Table 10.** Working conditions.

| Property | Quantity | Units |
| --- | --- | --- |
| Heating rate | 100 | $\frac{°C}{min}$ |
| Maximum temperature | 1000 | °C |
| Minimum Pressure | 4 | bar |

The classification of requirements is based on the nature of each one. Geometric requirements, such as height, thickness, diameter, length, width, and so on, belong to the design shape. Kinematic requirements refer to the kind of movement, direction of movement, speed, and acceleration. Force requirements include the force direction, force magnitude, frequency, weight, load, deformation, elasticity, inertial force, and resonance.

Energy requirements include aspects like the output, efficiency, loss, friction, ventilation, pressure, temperature, heating, cooling, and capacity. Material requirements cover physical or chemical properties of materials, auxiliary materials, and predefined materials, such as food-grade. Each requirement belongs to a classification, as shown below:

Classification of requirements:

- Geometric (G)
- Kinematic (K)
- Force(F)
- Energy (E)
- Material (M)
- Signals (Sig)
- Safety (S)
- Ergonomics (Ergo)
- Production (P)
- Quality Control (QC)
- Assembly (As)
- Transportation (T)
- Operation (O)
- Maintenance (Man)
- Recycling (R)
- Costs (C)
- Timelines (H)

8.2.2. Technical requirements

The technical requierements are defined in order to propose the solution. In addition, the Table 11 shows a set of values to establish an importance value for each requirement.

**Table 11.** Value definition to assign a level of importance.

| Number | Requirement | Desire or Demand | Classification |
|--------|-------------|------------------|----------------|
| 1 | Importance for insufficient design | Desire | D1 |
| 2 | Importance for low design | Desire | D2 |
| 3 | Importance for medium design | Desire | D3 |
| 4 | Importance for acceptable design | Demand | R1 |
| 5 | Importance for essential design | Demand | R2 |

In Table 12, the requirements are classified by their level of importance. Technical features and operation conditions are considered in Table 12.

**Table 12.** Requirements list under the methodology proposed.

| Importance Level | Classification | Requirements | Value |
|------------------|----------------|--------------|-------|
| D3 | G | Length | 490 mm |
| D3 | G | Diameter | 150 mm |
| R2 | G | Thickness | 8 mm |
| R1 | E | Maximum working temperature | 1000 °C |
| D2 | E | Minimum design pressure | 4 bar |
| R2 | E | External heating | List |
| R1 | E | Heating rate | $100 \, \frac{°C}{min}$ |
| R1 | M | Low corrosion | List |

**Table 12.** *Cont.*

| Importance Level | Classification | Requirements | Value |
|:---:|:---:|:---:|:---:|
| R1 | M | Limited rigidity with minimum mass | List |
| D3 | M | Performance before rupture | List |
| D3 | M | Limited resistance to minimum mass | List |
| D2 | S | Safety system | List |
| R1 | P | Adherence to tolerances | List |
| R1 | QC | Application of ASME code | Graph |

8.2.3. Conceptual Design

The conceptual design is based on creative ideation; after that, principles, laws, operating codes, governing equations, and general operations are implemented to refine the conceptual design. Since the final product design will be used in the pyrolysis process, it is also important to consider the principles of operation and work. During the reactor operation, the degradation of biomass in the absence of oxygen until the reactor reaches 600 °C is one of the main features to be considered during the process of design conceptualization. According to [83], this temperature is considered the pyrolytic temperature. If this process is carried out in a controlled manner, the products that can be obtained include liquids, ashes, and non-condensable gases. Finally, the general process is described in Equation (3) and it is shown as follows:

$$C_nH_mO_p(Biomass) = \sum_{Tar}C_xH_yO_z + \sum_{gas}C_aH_bO_c + C \tag{3}$$

The conceptual design is formulated through a creative process that considers the theoretical background of pressure vessel devices and standards. In Figure 8, an external burner heating is proposed as a concept to solve the problem provided by the customer. The concept uses a fixed-bed reactor that is heated by liquefied petroleum gas (LPG). The heat produced by the combustion is transferred by the principle of convection between the flame and the reactor. The temperature rises from the bottom to the top by conduction in the reactor.

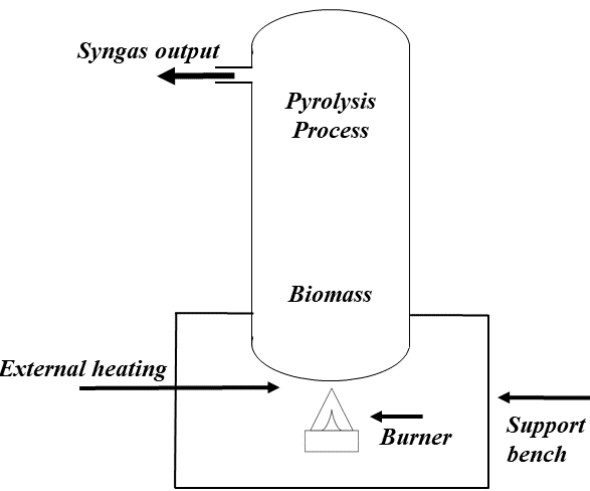

**Figure 8.** External burner heating.

In Figure 9, a concept using external heating by a coil is shown. The concept considers the use of a fixed-bed reactor. It is heated by a coil outside the reactor. In this concept, a steam flow is generated by the external coil heating, then it passes through the entire reactor surface; as a result, this design provides uniform heating to the reactor. In this case,

the heating transfer is carried out by the principle of convection between the steam coil and the reactor surface.

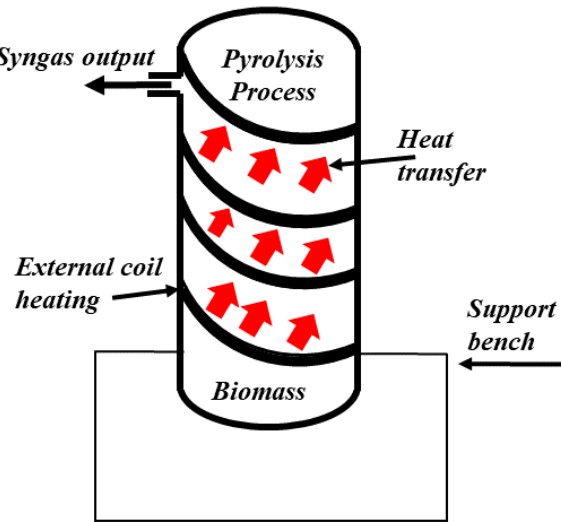

**Figure 9.** External coil heating.

In Figure 10, a concept based on a jacketed reactor is proposed. In this case, the heating of the jacketed reactor is carried out by a burner flame inlet opening. This configuration is used to maintain a uniform temperature in the reactor. The heat transfer is also done by the convection principle between the flame and the internal reactor.

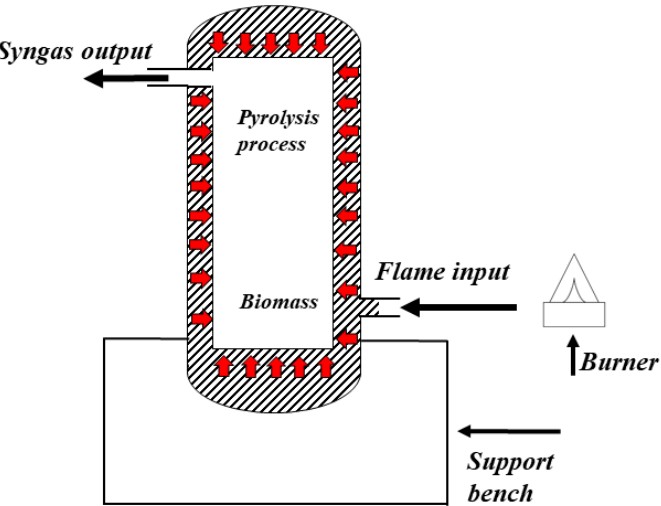

**Figure 10.** Jacketed reactor.

8.2.4. Prototype Design

The ASME code is an international standard for the design, construction, and manufacturing of pressure vessels. Hence, the design of the reactor is carried out according to the ASME code [84].

The main geometrical specification of the pressure vessel is the thickness of the vessel shell. The vessel shell must contain the maximum internal working pressure to ensure the safe operation of the vessel. The computation of the pressure vessel geometry is performed according to ASME code specifications. Equation (4) is used to compute the thickness of the vessel shell as follows:

$$t = \frac{D}{2}\left(exp\left[\frac{P}{SE}\right] - 1\right) \tag{4}$$

$t$ = Minimum  required thickness of a shell.
$D$ = Inside diameter of a shell or head.
$P$ = Internal design pressure.
$S$ = Allowable stress value.
$E$ = Weld joint factor.

Then, a cylindrical pressure vessel with ellipsoidal heads is computed, so the design equations (Equations (5) and (6)) are established to obtain the inner radius of the knuckle and the inner radius of the crown of the head, respectively [84].

$$r = D\left(\frac{0.5}{k} - 0.8\right) \tag{5}$$

$r$ = Inside  knuckle  radius  used  in  torispherical  head  calculation.
$D$ = Inside diameter of a shell or head.
$k$ = Angular  constant.

$$L = D(0.44k + 0.02) \tag{6}$$

$L$ = Inside crown  radius of a torispherical head.
$D$ = Inside diameter of a shell or head.
$k$ = Angular  constant.

To obtain the angular constant $k$:

$$k = \frac{D}{2h} \tag{7}$$

$k$ = Angular  constant.
$D$ = Inside diameter of a shell or head.
$h$ = Height of the ellipsoidal head measured to the inside surface.

The ASME code specifies that the angular constant must oscillate within the range of 1.7 and 2.2 for ellipsoidal heads. In order to compute this feature, the following equations must be satisfied:

$$1.7 \leq k \leq 2.2 \tag{8}$$

To obtain the head ratios:

$$0.7 \leq \frac{L}{D} \leq 1.0 \tag{9}$$

$$\frac{r}{D} \geq 0.06 \tag{10}$$

$$20 \leq \frac{L}{r} \leq 2000 \tag{11}$$

Equations (12)–(16) are used to compute the angles used in the ellipsoidal head.

$$\beta_{th} = \arccos\left[\frac{0.5D - r}{L - r}\right] \tag{12}$$

$\beta_{th}$ = Angle used in the torispherical head.
$D$ = Inside diameter of a shell or head.
$r$ = Inside knuckle radius used in torispherical head.
$L$ = Inside crown  radius of a torispherical head.

$$\phi_{th} = \frac{\sqrt{Lt}}{r} \tag{13}$$

$\phi_{th}$ = Angle used in the torispherical head.
$L$ = Inside crown  radius of a torispherical head.
$r$ = Inside knuckle radius used in torispherical head.
$t$ = Minimum required thickness of a shell.

$$R_{th} = \frac{0.5D - r}{\cos[\beta_{th} - \phi_{th}]} + r \tag{14}$$

$R_{th}$ = Radius used in the torispherical head.
$D$ = Inside diameter of a shell or head.
$r$ = Inside knuckle radius used in torispherical head.
$\beta_{th}$ = Angle used in the torispherical head.
$\phi_{th}$ = Angle used in the torispherical head.

$$C_1 = 0.692\left(0.692\frac{r}{D}\right) + 0.625 \tag{15}$$

$C_1$ = Angle used in the torispherical head.
$r$ = Inside knuckle radius used in torispherical head.
$D$ = Inside diameter of a shell or head.

$$C_2 = 1.46 - 2.6\left(0.692\frac{r}{D}\right) \tag{16}$$

$C_2$ = Angle used in the torispherical head.
$r$ = Inside knuckle radius used in torispherical head.
$D$ = Inside diameter of a shell or head.

Equation (17) is used to obtain the expected internal pressure since it could be the cause of the elastic buckling of the knuckle.

$$P_{eth} = \frac{C_1 E_T t^2}{C_2 R_{th}\left(\frac{R_{th}}{2r} - 1\right)} \tag{17}$$

$P_{eth}$ = Internal pressure expected to produce elastic buckling of the knuckle in a torispherical head.
$C_1$ = Angle used in the torispherical head.
$E_T$ = Modulus of elasticity at maximum design temperature.
$D$ = Inside diameter of a shell or head.
$t$ = Minimum required thickness of a shell.
$C_2$ = Angle used in the torispherical head.
$R_{th}$ = Radius used in the torispherical head.
$r$ = Inside knuckle radius used in torispherical head.

Equation (18) is used to compute the internal pressure. This pressure will result in maximum stress at the knuckle, equal to the material's yield strength.

$$P_y = \frac{C_3 t}{C_2 R_{th}\left(\frac{R_{th}}{2r} - 1\right)} \tag{18}$$

$P_y$ = Internal pressure expected to result in a maximum stress equal to the material yield strength in a torispherical head.
$C_3$ = Strength parameter used in the torispherical head.
$t$ = Minimum required thickness of a shell.
$C_2$ = Angle used in the torispherical head.
$R_{th}$ = Radius used in the torispherical head.
$r$ = Inside knuckle radius used in torispherical head.

In order to obtain the parameter $C_3$, there are different cases to determine it, which are defined according to the following statements: if the allowable stress at the calculation temperature is governed by time-independent properties, then $C_3$ is the material's yield strength at the calculation temperature. If the allowable stress at the calculation temperature is governed by time-dependent properties, then $C_3$ is determined by the following: if the allowable stress is established based on the 90% yield criterion, then $C_3$ is the allowable stress of the material at the calculation temperature multiplied by $C_3 = 1.1S$. Finally, if the allowable stress is established based on the 67% yield criterion, then $C_3$ is the allowable stress of the material at the calculation temperature multiplied by $C_3 = 1.5S$.

On the other hand, Equation (19) is used to compute the internal pressure expected to cause knuckle buckling.

$$P_{ck} = \left( \frac{0.77508G - 0.20354G^2 + 0.019274G^3}{1 + 0.19014G - 0.089534G^2 + 0.0093965G^3} \right) P_y \tag{19}$$

$P_{ck}$ = Internal pressure expected to result in buckling failure of the knuckle.
$G$ = Constant used in the torispherical head.
$P_y$ = Internal pressure expected to result in a maximum stress equal to the material yield strength in a torispherical head.

The constant, $G$, is computed by Equation (20), which defines the torispherical head:

$$G = \frac{P_{eth}}{P_y} \tag{20}$$

$G$ = Constant used in the torispherical head.
$P_{eth}$ = Internal pressure expected to produce elastic buckling of the knuckle in a torispherical head.
$P_y$ = Internal pressure expected to result in a maximum stress equal to the material yield strength in a torispherical head.

The computation of the allowable pressure based on knuckle buckling is given by Equation (21), as follows:

$$P_{ak} = \frac{P_{ck}}{1.5} \tag{21}$$

$P_{ak}$ = Allowable internal pressure of a torispherical head based on a buckling failure of the knuckle.
$P_{ck}$ = Internal pressure expected to result in buckling failure of the knuckle.

Similarly, the computation of the allowable pressure based on the rupture of the crown is given by Equation (22), as follows:

$$P_{ac} = \frac{2SE}{\frac{L}{t} + 0.5} \tag{22}$$

$P_{ac}$ = Allowable internal pressure of a torispherical head based on the rupture of the crown.
$S$ = Allowable stress value.
$E$ = Weld joint factor.
$L$ = Inside crown radius of a torispherical head.
$t$ = Minimum required thickness of a shell.

The computation of the maximum permissible internal pressure is given by Equation (23), as follows:

$$P_a = min[P_{ak}, P_{ac}] \tag{23}$$

$P_a$ = Maximum allowable internal pressure of a torispherical head.
$P_{ak}$ = Allowable internal pressure of a torispherical head based on a buckling failure of the knuckle.
$P_{ac}$ = Allowable internal pressure of a torispherical head based on the rupture of the crown.

The aforementioned theoretical background allows one to obtain the maximum permissible pressure for the reactor design. On the other side, the material selection is conducted according to the steps shown in the flowchart in Figure 5.

In Figures 11–15 the material properties are shown; they should be considered to create a pressure vessel device [60,85]. Each aforementioned figure presents the metals and alloys in colors; each color determines the characteristics of the materials. The commercial steels are blue, alloys with titanium are purple, and yellow and dark green colors denote the ceramic materials.

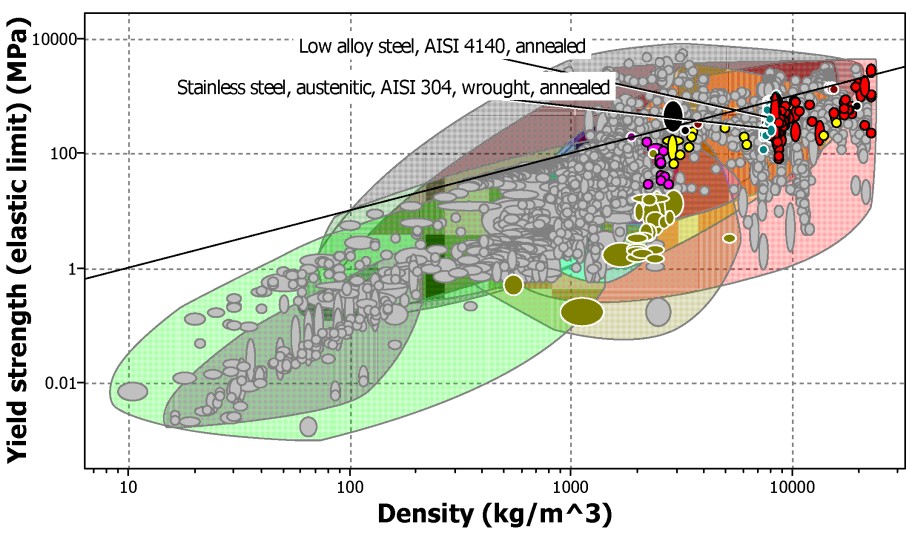

**Figure 11.** Graph of Young's modulus versus the density of materials $\frac{E}{\rho}$.

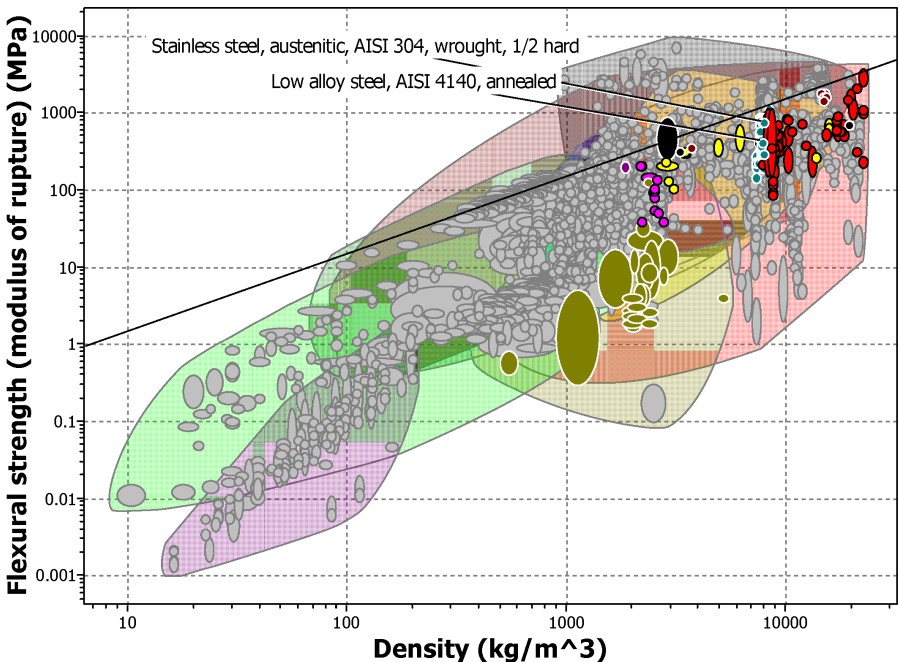

**Figure 12.** Graph of flexural strength versus the density of materials $\frac{\sigma_f}{\rho}$.

In accordance with the definition of the technical requirements by the designer, the material properties should consider the following parameters: the limit of rigidity at the minimum mass. Thus, the relation $\frac{E}{\rho}$ should be considered in order to find the solution to this requirement. In Figure 11, the relation $\frac{E}{\rho}$ in a graph is shown. This graph plots Young's modulus $[E]$ versus the density of the materials $[\rho]$.

In addition, another technical requirement of the design pertains to the next parameter: the limit of resistance at the minimum mass. For this requirement, the relation $\frac{\sigma_f}{\rho}$ should be used; it is shown in Figure 12. This graph plots the relation between flexural strength and the density of the materials.

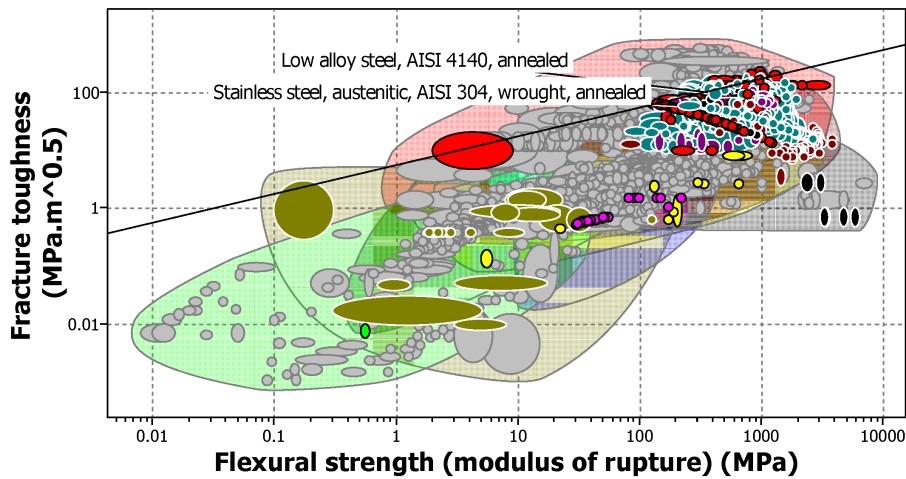

**Figure 13.** Graph of the fracture toughness versus the flexural strength of materials $\frac{K_{1c}}{\sigma_f}$.

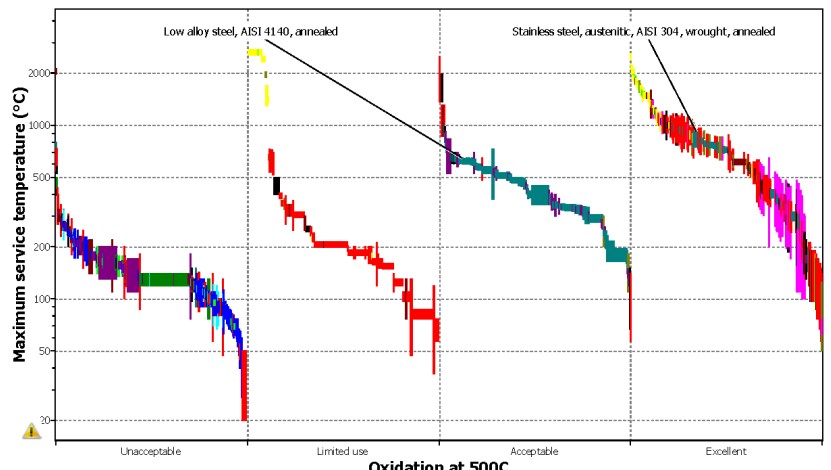

**Figure 14.** Graph of maximum service temperature and oxidation at 500 °C.

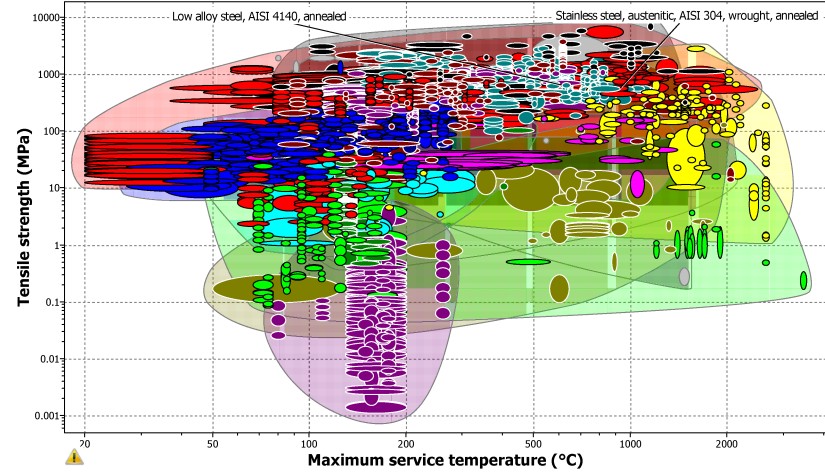

**Figure 15.** Graph Tensile of strength and maximum service temperature.

In Figure 13, the relation $\frac{K_{1c}}{\sigma_f}$ is shown; it is used to select materials based on the fracture toughness versus flexural strength of the materials. The index selected in the graph is used to determine the damage tolerance of the design. This is a technical requirement regarding the material performance before rupture is observed.

Conversely, the correlation between the maximum service temperature versus the oxidation at 500 °C is shown in Figure 14. This requirement is related to the durability against the oxidation of the material at high temperatures. The material selected should present a low resistance to oxidation, and the material should be durable.

The change in tensile strength at high temperatures is one of the parameters to study in the materials. This parameter is key for designers since the tensile strength changes with respect to temperature and tends to decrease as the temperature increases. The tensile strength against the maximum service temperature is shown in Figure 15.

After analyzing Figures 11–15 under several temperature conditions, the first consideration for the selection of material is the use of commercial steels, as they present the following properties. From Figure 11, the steels considered are those with a density between 7500 and 8000 $\frac{kg}{m^3}$. According to the figure, the selected material presents a minimum mass with respect to stable stiffness. Moreover, in Figure 12, steels within a density range of 7500–8000 $\frac{kg}{m^3}$ and with a modulus of rupture of 200–500 MPa are considered, as these steels present limited strength with minimum mass.

In addition, Figure 13 shows materials that must present acceptable performance before rupture. The considered material can absorb energy during their plastic deformation, and their properties should be within the range of a flexural strength of 200–500 MPa and fracture toughness of 50–100 MPa $m^{0.5}$.

On the other hand, Figure 14 presents materials that have excellent or acceptable resistance to oxidation between 500 °C and 1000 °C, which are the operating conditions of the pressure vessel. From Figure 15, the selected materials must present a tensile strength greater than 500 MPa at 500 °C, as some materials change their properties at high temperatures.

The selected materials are AISI 304 and AISI 4140. AISI 304 is an annealed wrought and austenitic stainless steel, and AISI 4140 is an annealed low alloy steel. Since the customer requirement indicates that the minimum working temperature is 600 °C and the maximum service temperature is 1200 °C, the range of selection is reduced to the materials indicated in blue.

8.2.5. Refine Details

In this section of the material-focused design methodology, the minimum design pressures of the pressure vessel are obtained following the conditions of the ASME code for the calculation of this type of device. These pressures are shown in Table 13. This approach enables the verification of the necessary pressure vessel and the implementation of safety measures during the operation of the process, including the selection of relief valves at 7 MPa as a safety factor.

**Table 13.** Operating pressures.

| Condition | Symbol | Value | Units |
| --- | --- | --- | --- |
| Allowable pressure based on knuckle buckling failure | $P_{ak}$ | 8.35 | MPa |
| Allowable pressure based on crown rupture | $P_{ac}$ | 31.5 | MPa |
| Maximum allowable pressure | $P_a$ | 8.35 | MPa |

Two different types of materials are selected, which can be used for the design of a pressure vessel for the pyrolysis process. These two materials are commercial steels recurrently used in different types of processes. These materials were selected using the property graphs; their properties are shown in Table 14. These materials meet certain mechanical, thermal, and durability properties for the temperature conditions considered. Therefore, they are considered to have low corrosion during their time of use, meaning low maintenance during operations, minimal loss of thickness over time, and stability during

high temperatures. Two optimal materials, AISI 304 and AISI 4140, are selected as they primarily meet the temperature, pressure, and toughness specifications.

**Table 14.** Material properties [59].

| Material | Property | Value | Units |
|---|---|---|---|
| AISI 304 | Young's Modulus | 190–203 | GPa |
| | Failure Strength | 150–220 | MPa |
| | Tensile Strength | 510–620 | MPa |
| | Fracture Toughness | 55–71 | MPa m$^{1/2}$ |
| | Maximum Service Temperature | 750–925 | °C |
| | Durability at 500 °C Oxidation | Excellent | |
| AISI 4140 | Young's Modulus | 208–216 | GPa |
| | Failure Strength | 183–248 | MPa |
| | Tensile Strength | 595–720 | MPa |
| | Fracture Toughness | 76–120 | MPa m$^{1/2}$ |
| | Maximum Service Temperature | 613–650 | °C |
| | Durability at 500 °C Oxidation | Acceptable | |

Sustainability is a key concept that designers and producers should consider in order to manufacture new devices or products. For this reason, in the proposed case study, the selection of materials is carried out by considering at least some of the following aspects:

- The material selected is recyclable.
- The material presents a high resistance to corrosion.
- The material can withstand temperatures above 500 °C.
- The material selected should have an eco-indicator near 95 or 99.
- The $CO_2$ footprint during the first production is lower compared to existing products in the market.

The eco-indicators are numbers that express the total environmental impact of a process or product because all products and processes pollute. This is due to the fact that raw materials must be extracted, manufactured, distributed, assembled, and disposed of [86]. Hence, all stages of the product life cycle must be studied. The eco-indicator value for AISI 304 is 424 millipoints per kg, and for AISI 4140, it is 110 millipoints per kg, respectively [86–89]. The $CO_2$ footprint refers to the equivalent mass of greenhouse gases produced and released into the atmosphere as a result of the production of one kilogram of material. In the case of AISI 4140 steel, the equivalent mass ranges from 1.93 to 2.14 kg of $CO_2$ per kg of material, and for AISI 304, it ranges from 5.17 to 5.71 kg of $CO_2$ per kg of material [87,90]. Both materials can be recycled after fulfilling their operating cycles, according to [87]. Similarly, for these materials, the corrosion condition is well-known, as it presents low resistance to corrosion at temperatures above 500 °C [87,91–93].

*8.3. Quality Function Deployment*

The entire creative process of the QFD methodology for generating concept design is beyond the scope of this article, so the authors only present the part of obtaining technical requirements by this methodology, which is the main contribution of this article. The QFD methodology is implemented in the aforementioned case study, and then it is compared with the methodology proposed. The requirements for the same problem have been obtained by the QFD methodology, and they are shown in Table 15.

**Table 15.** Customer requirements.

| Requirement | Value |
|---|---|
| Maximum process temperature | 1000 °C |
| Heating rate | 100 $\frac{°C}{min}$ |
| Pyrolytic Temperature | 100 °C |
| Maximum working pressure | 6 bar |
| Minimum design pressure | 4 bar |
| High-temperature durable material | List |
| Compliance with international standards | List |
| Safety system | List |
| Low corrosion | List |
| Low vessel costs | List |
| Compliance with geometric dimensions and tolerances | List |

In order to verify the feasibility of the requirements obtained, the next step involves presenting them to the customer to ascertain if the requirement descriptions meet their needs [20,26,27,39]. Moreover, the requirements can regard the type of operation to be performed, the geometry, specific characteristics, or working conditions. Then, the customer's requirements are translated into technical requirements; this task is commonly carried out by a designer with a high amount of experience [20,26,27,39]. The customer requirements are described in Table 15 in order to build the quality house. According to the QFD methodology, it is important to assign a level of importance to each requirement. Hence, the value scale is defined on a scale that ranges from 1 to 5, where value 1 is the least important of all, and 5 is the most important requirement, as shown in Table 16.

**Table 16.** Customer requirements by QFD methodology.

| Row | Relationship Value | Relative Weight | Weight/Importance | Demanded Quality |
|---|---|---|---|---|
| 1 | 9 | 15.4 | 4.00 | Maximum working temperature |
| 2 | 9 | 7.7 | 2.00 | Minimum design pressure |
| 3 | 9 | 15.4 | 4.00 | Low corrosion |
| 4 | 9 | 7.7 | 2.00 | Safety system |
| 5 | 9 | 15.4 | 4.00 | Adherence to tolerances |
| 6 | 9 | 11.5 | 3.00 | Durable |
| 7 | 9 | 15.4 | 4.00 | Application of ASME code |
| 8 | 9 | 11.5 | 3.00 | Low costs |

Afterward, each customer requirement is assigned a level of importance. These requirements are then translated into technical requirements, which form the basis for designing the product. The QFD methodology suggests carrying out a benchmarking process to compare and understand how competitors have solved the same problem. An evaluation is then conducted to identify the most and least important requirements. The top of the house is built with an evaluation related to technical requirements, as more than one may have a relationship with others. As a result of this process, the quality house for the product design is completed and is shown in Figure 16.

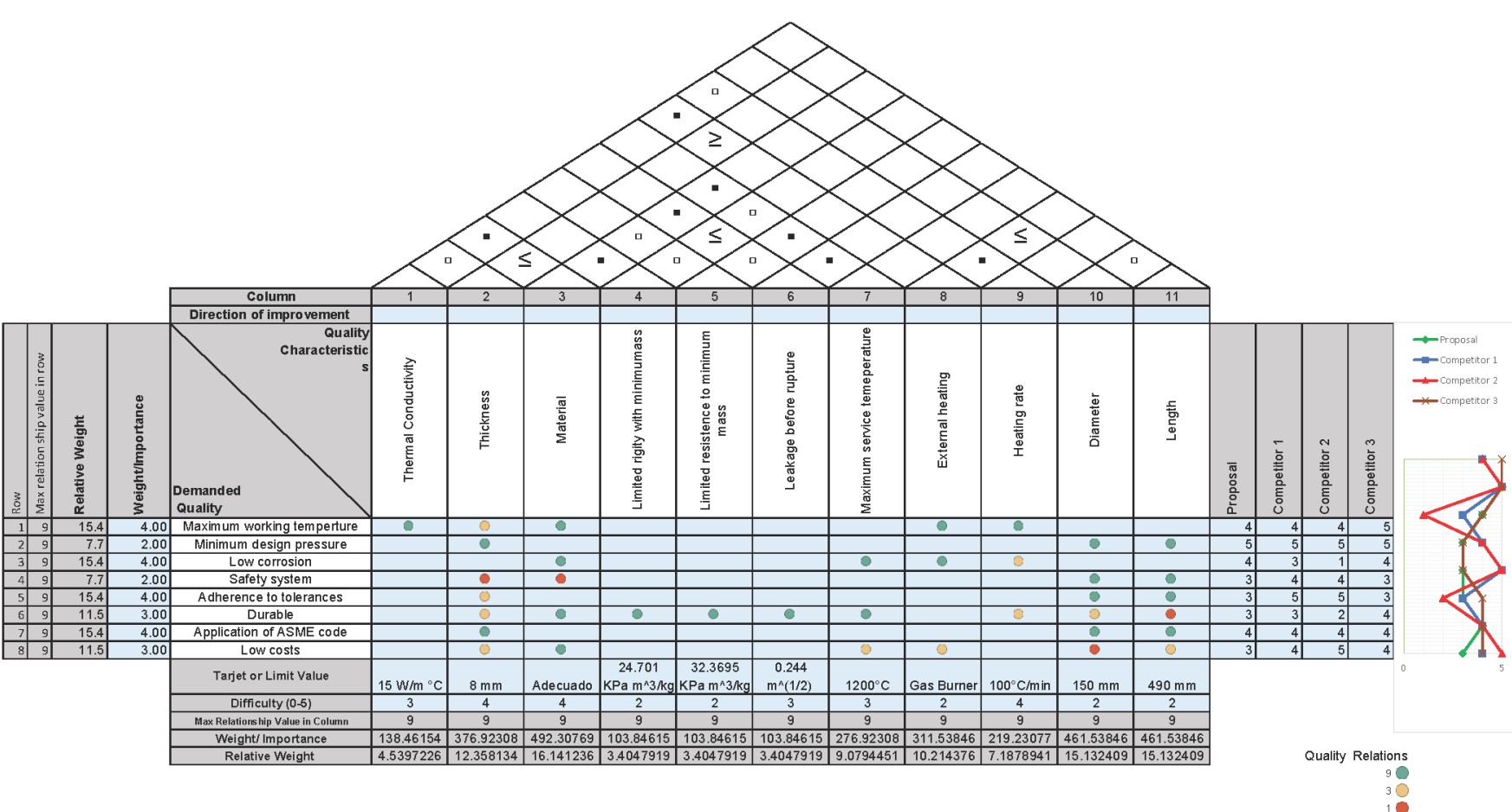

**Figure 16.** Case study implemented by QFD methodology.

Finally, the technical requirements for the design of the pressure vessel are obtained by the QFD methodology. These technical requirements include geometry, heating range, pyrolytic temperature of operation, ranges of pressure during operation, security features, corrosion rates, and costs. The technical requirements obtained by the QFD methodology are subject to an evaluation, which is carried out after the concept design. The concept design is a creative process intended to propose a suitable concept to meet the customer's needs. After an evaluation of the concept design, a proof of concept is carried out; finally, the final requirements and concepts are selected. These final requirements are then converted into a solution for the customer.

## 9. Discussion

The material-focused design methodology proposes a design that focuses not only on customer requirements but also on the materials used to build the product. The first step in applying the proposed methodology is to clarify the main idea, as it is very important to satisfy the client's requirements. Then, a description of the requirements is carried out to define which are more important to the customer and the approach to the problem. The next step is to establish a level of importance by assigning a value to each requirement. As a result, the main requirements obtained were the wall thickness of the vessel, the external heating of the vessel, the maximum working temperature of the vessel (which establishes one of the first requirements for the material), the minimum design pressure for the vessel, the heating rate to increase the temperature of the vessel, and the low corrosion of the material. The obtained requirements can be translated into technical features by the support of standards with a metric and value. This feature is important to reduce the interactive procedure carried out by the QFD methodology. The use of standards during the process of obtaining customer requirements minimizes deviations and errors in the conceptual design, as shown in the case study. Similarly, the use of correct material selection can be established from the beginning of the project to avoid spending time on design iterations that result in constraints. The suitable selection of material is one of the main requirements that customers need; hence, standards serve as tools that help designers in satisfying this need. In comparison with the proposed methodology, the QFD implementation presents a recursive process to obtain technical requirements. The translation from customer requirements to technical requirements depends on the designer's experience, as the QFD methodology implements a step-by-step approach but does not indicate technical standards. This activity could be an interactive process followed by an inexperienced designer until the correct design is reached. For example, the designer could have a recursive process to establish technical requirements such as the selection of the material, the material's thermal conductivity, the maximum operating temperature, the durability of the material before corrosion, the density of the material, the modulus of elasticity, the resistance to bending, and the resistance to fracture. These parameters are properties of the materials which, depending on the application, could be obligatory. In the case of a pressure vessel, there are a wide range of standards to design it; hence, it is important to have a methodology that guides designers to develop a well-conceived solution. According to the proposed methodology, the type of material and geometry are well-established from the beginning.

The application of both methodologies to the same case study highlights the advantages of the proposed methodology since the QFD methodology presents an interactive process throughout its entirety.

## 10. Conclusions

In this work, a novel design methodology was proposed to address multidisciplinary customer requirements. The proposed methodology includes the use of standards for the correct selection of materials. Designing a pressure vessel for the pyrolysis process was proposed as a case study. The case study was evaluated using both the QFD methodology and the Material-focused Methodology. The results of the QFD implementation show

an interactive process is needed to refine the customers' requirements before finding a suitable solution. This interactive process requires more time during the design process. Conversely, the proposed methodology shows a suitable selection of materials from the beginning of establishing requirements, as demonstrated in the case study. This results in a more agile design process for designers and customers. In addition to the multidisciplinary requirements and standards used, the methodology includes sustainable considerations to deal with the environmental requirements for producing new devices around the world.

**Author Contributions:** Investigation, formal analysis, writing—original draft, E.A.M.G., S.I.O. and L.A.S.; review & editing, G.J.G.P. and J.d.J.R. All authors have read and agreed to the published version of the manuscript.

**Funding:** Authors thank the Instituto Politécnico Nacional, Secretaría de Investigación y Posgrado, Comisión de Operación y Fomento de Actividades Académicas for their help in this research.

**Data Availability Statement:** Date are contained with the article.

**Conflicts of Interest:** The authors declare no conflict of interest.

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
