# Peer review of "Exploring a Material-Focused Design Methodology: An Innovative Approach to Pressure Vessel Design"

_machines, doi:10.3390/machines12010081_

Round 1

Reviewer 1 Report

Comments and Suggestions for Authors

This study reviewed the methods previously used for guiding the design of industrial production, analyzed their advantages and disadvantages, and proposed a multidisciplinary methodology that blended multiple methodologies. Moreover, the author studied the pyrolysis process, and a product that met the requirements with this innovative methodology was obtained. Overall, this study made a contribution to sustainable development, material utilization, and the efficiency of production, and could better address the needs of the customer. However, several points need to be improved. Specific comments follow.

Abstract

p.1, line 8 - 12: This section seems to illustrate the problem that this study aims to solve, but it is kind of ambiguous and insufficient. Suggest directly proposing the concrete shortcomings of existing design methodology.

Introduction

p.1, line 16: The word “Introducción” should be “introduction”

p.2, line 51: Please note that the first word in the first line of a paragraph should have blank spaces in the front.

p.2, line 55, 56: Please note the logic of your expression. The previous paragraph mentions the existing design methodology, but did not clearly indicates the specific reasons for the lack of a suitable methodology for pressure vessels. The use of ‘Thus’ here seems illogical. Please provide more detailed explanation

p.2, line 56 - 58: It appears that Ferrer’s work is not related to the topic and please explain the purpose of citing his work here.

Methodology QFD

p.2, line 70: The word “Metodología” should be “Methodology”

Methodology Comparison

p.11, line 219: Suggest deleting the sentence “Which methodology is compatible with my design?” because the interrogative sentence appears not very rigorous and scientific in the paper.

Material-focused Design Methodology

p.12, line 259 - 263: The proposed method here is very abrupt, as the previous content is all about independent methodology but doesn’t mention the existing work related to the multidisciplinary methodology, making the new methodology lack of scientific basis and reliability. Please investigate other relevant work

p.13, line 264 - 296: This methodology lacks experimental verification and feasibility. The following contents show only one case with required product to verify the multidisciplinary methodology, and there’s no comparison between the results of this case and that of the other methodologies. Suggest increasing the cases for study and comparing them with the previously mentioned independent methodology.

Case Study

p.20: Suggest attaching a concise explanation below to all properties graph in this section, including the constituent elements of the graph and the meanings of each region and color.

Results

p.22, line 409: The sentence “In the design process, the aim is to determine specifications” should be “The aim of the design process is to determine specifications …”

p.23, line 416: The word “CConsidering” should be “Considering”

Comments on the Quality of English Language

N/A

Reviewer 2 Report

Comments and Suggestions for Authors

This work presents a methodology that incorporates multidisciplinary into the design of industrial products. Finally, the methodology is applied to a case study, resulting in a product designed under a multidisciplinary approach to meet the customer needs.The design of the new methodology generates a process through which it is possible to identify the needs of a new product that must meet certain mechanical, thermal, and electrical material properties or specific conditions that any material would not fulfill. The paper is well written and well designed. This is  a kind of perfect works. The results are well interpreted. I recommend its publication after minor revision. The only comment is about Eqs. (23), (24). I think, no need to give numbers for these expressions, they can be mentioned within the text.

Reviewer 3 Report

Comments and Suggestions for Authors

1) The abstract must briefly contain some new design methods proposed in the work, as well as concrete advantages in relation to the old known methods. Some aspects presented in lines 63-69 should be briefly inserted in the abstract: "In section 2, the Quality Function Deployment (QFD) methodology is presented, in section 3, the Theory of Inventive Problem Solving methodology is presented, in section 4, Ashby’s Materials Selection methodology, in section 5, the Systematic Approach methodology is presented, in section 6, a comparison of the aforementioned methodologies is presented, in section 7, the Material-Focused Design methodology, in section 8, the case study is presented, in section 9, the obtained results are presented, and in section 10, the conclusions are presented."
2) The next sentence seems unfinished: "There are some very clear examples, such as gears which are the mean for the transformation and utilization of mechanical energy, steam generators for the transformation of thermal energy into electrical energy, the design of vehicles and trains for transportation of people, the design of electrical devices to enhance home comfort, the design of airplanes or spacecraft, the design of robots for hazardous tasks in the industry, and recently for medical assistance in patients, the design of medical use devices, etc." (rows 26-31).
3) You remember and exemplify the basic concept of "fundamental laws or working principles", but this concept mentioned in the paper must be explained at its first appearance in work and possibly quoted: "Currently, these devices are designed using methodologies that use basic concepts such as fundamental laws or working principles. (lines 31-33).”
4) Before Table 1, briefly present all the aspects included in the table as well as the citations in it.
5) The data in Figures 1 and 2 as well as the related text that follows them (lines 106-130) require citations.
6) Discuss Tables 2-4 in more detail.
7) Discuss Figures 4-5 in more detail.
8) Discuss Tables 5-7 in more detail.
9) Discuss Figures 8-10 in more detail.
10) Discuss equations 3-23 in more detail.
11) Discuss Figures 11-15 in more detail.
12) ?  row 416: "(C)Considering".
13) Discuss the "Results" section in more detail.
14) The conclusions section must be redone completely, in detail, and highlighting the novelties brought by the work.

Comments on the Quality of English Language

 Moderate editing of English language required

Round 2

Reviewer 1 Report

Comments and Suggestions for Authors

The authors solved my concerns and the revised paper is suitable for publication.

Reviewer 3 Report

Comments and Suggestions for Authors

 Accept in present form

Comments on the Quality of English Language

 Minor editing of the English language required